# A Geomagnetic/Odometry Integrated Localization Method for Differential Robot Using Real-Time Sequential Particle Filter

**DOI:** 10.3390/s24072120

**Published:** 2024-03-26

**Authors:** Qinghua Luo, Mutong Yu, Xiaozhen Yan, Zhiquan Zhou, Chenxu Wang, Boyuan Liu

**Affiliations:** 1School of Information Science and Engineering, Harbin Institute of Technology at Weihai, Weihai 264209, China; luoqinghua081519@163.com (Q.L.); 23b905030@stu.hit.edu.cn (M.Y.); zzq@hitwh.edu.cn (Z.Z.); wangchenxu@hit.edu.cn (C.W.); 23s130341@stu.hit.edu.cn (B.L.); 2Shandong Institute of Shipbuilding Technology, Ltd., Weihai 264209, China

**Keywords:** differential robot, geomagnetic matching navigation, odometry, particle filter

## Abstract

Geomagnetic matching navigation is extensively utilized for localization and navigation of autonomous robots and vehicles owing to its advantages such as low cost, wide-area coverage, and no cumulative errors. However, due to the influence of magnetometer measurement noise, geomagnetic localization algorithms based on single-point particle filters may encounter mismatches during continuous operation, consequently limiting their long-range localization performance. To address this issue, this paper proposes a real-time sequential particle filter-based geomagnetic localization method. Firstly, this method mitigates the impact of noise during continuous operation while ensuring real-time performance by performing real-time sequential particle filtering. Then, it enhances the long-range positioning accuracy of the method by rectifying the trajectory shape of the odometry through odometry calibration parameters. Finally, by performing secondary matching on the preliminary matching results via the MAGCOM algorithm, the positioning error of the method is further minimized. Experimental results show that the proposed method has higher positioning accuracy compared to related algorithms, resulting in reductions of over 28.58%, 37.11%, and 0.77% in RMSE, max error, and error at the end, respectively.

## 1. Introduction

In autonomous robot navigation, attaining cost-effective, highly precise, and real-time localization is imperative. A differential robot [1] is a type of autonomous robot based on a differential drive, which means controlling the speed difference between two wheels to achieve direction control and turning. This simple and efficient design makes differential robots popular for various applications. Wheel odometry [2] is one of the commonly used navigation methods for differential robots, which serves as a sensor for measuring the displacement and direction of a moving vehicle. It uses pulse count data generated by encoders to measure the rotation angle of wheels [3] and computes the vehicle’s motion through established motion models. While wheel odometry presents advantages such as affordability and real-time performance, it is susceptible to issues of error accumulation [4]. The precision of measurement can be affected by tire slippage, tire deformation, uneven ground surfaces, and other factors. Consequently, in practical applications, the integration of wheel odometry with other sensors is often employed to enhance the accuracy and robustness of localization.

Geomagnetic-aided navigation (GMN) [5] is a technique that utilizes Earth’s magnetic field information for navigation. It supports other navigation systems, such as the inertial navigation system (INS), by correcting the provided position and orientation. This is achieved by matching the measurement of the geomagnetic field intensity at the current position with the geomagnetic reference map of nearby regions. Consequently, this method enhances the navigation accuracy of robots and vehicles. Compared to other localization and navigation methods, GMN offers advantages such as low cost, wide-area coverage, and no cumulative errors [6,7]. As a result, it has been widely used in various fields. Recognizing the similarities between odometry and INS regarding high short-term positioning accuracy and the presence of cumulative errors over long distances [8], integrating GMN with wheel odometry offers an effective means to correct the accumulated errors in odometry. This integration facilitates long-distance, low-cost, high-precision real-time localization and navigation. The commonly used methods for geomagnetic-assisted navigation include geomagnetic filtering and geomagnetic matching.

Geomagnetic filtering is a real-time method for localization and navigation that analyzes one data point at a time. It addresses the accumulation of positioning errors by refining the current position provided by INS, odometry, and other systems through filtering. A Kalman filter [9,10] is one of the commonly used geomagnetic filtering algorithms. In 2014, reference [11] demonstrated the feasibility of the Sandia inertial terrain-aided navigation algorithm based on the Kalman filter for geomagnetic/INS integrated navigation. The observation of this algorithm relies on a linearized geomagnetic field model. However, geomagnetic models exhibit highly nonlinear characteristics, and the accuracy of the algorithm is affected by the linearization method. In comparison to the Kalman filter, the particle filter [12] offers better advantages in processing nonlinear geomagnetic data due to its good performance in addressing nonlinear and non-Gaussian estimation problems. In particle filtering, the state of a dynamic system is approximated by a set of weighted particles, with each particle representing a possible state and the associated weight representing the likelihood of that state being true. In 2016, reference [13] demonstrated the feasibility of the geomagnetic particle filtering algorithm for indoor pedestrian localization. In 2018, the geomagnetic particle filtering algorithm was used to locate autonomous surface vehicles, achieving better results compared to dead reckoning [14]. In 2020, Quintas et al. compared the effects of the extended Kalman filter, unscented Kalman filter, and particle filter in autonomous underwater vehicle navigation, demonstrating the better robustness of the particle filter in geomagnetic navigation [15]. In 2022, Lingfeng et al. optimized the geomagnetic particle filter using the firefly algorithm, reducing the problem of particle impoverishment and degradation [16]. In 2023, Benjamin proposed the use of geomagnetic particle filtering for indoor positioning of differential robots, simultaneously calibrating the magnetometers to effectively reduce the position and orientation errors [17]. However, such algorithms are prone to significant errors and even divergence in situations with strong geomagnetic noise. In order to avoid this situation, in 2023, Huapeng et al. used a particle filter for underwater positioning and introduced a distance interval between each execution of the algorithm to reduce errors caused by magnetometer measurement noise [18]. However, the long interval distance between the algorithm executions limits its ability to correct angular error, leading to a decline in positioning accuracy over time due to accumulated angular errors.

The geomagnetic matching method reduces the errors of navigation systems such as INS and odometry by matching the geomagnetic values and relative positions on a trajectory with the reference map. The commonly used geomagnetic matching methods include geomagnetic contour matching algorithms (MAGCOM) [19], iterative closest contours point (ICCP) [20], intelligent optimization algorithms [21], and neural networks [7]. In 2018, Xiao et al. proposed an improved ICCP algorithm that dynamically selects the appropriate matching length and matching points [20]. In the same year, Zhuo et al. proposed a geomagnetic vector ICCP algorithm based on searching the principle of trusted point sets, which improves the reliability of positioning compared to scalar matching [22]. In 2020, Chen et al. compared the applicability of MAGCOM, ICCP, and Sandia inertial magnetic aided navigation (SIMAN) algorithms in GMN for a supersonic aircraft [23]. In the same year, Wang et al. improved the geomagnetic matching algorithm based on particle swarm optimization (PSO) by using redundant information from geomagnetic measurements to constrain the particles, thus enhancing the algorithm’s noise resistance capability [24]. In 2022, Xu et al. proposed a combination of a PSO and ICCP algorithm to reduce the impact of initial errors on the ICCP [25]. In the same year, Jin et al. proposed an ICCP algorithm based on three reference maps of geomagnetic field vector data and demonstrated that using multiple reference maps in ICCP can further reduce the matching errors of single-component ICCP [26]. In 2023, Zhuo et al. used a probabilistic neural network for geomagnetic matching, which significantly reduced the probability of mismatch compared to traditional algorithms. These algorithms can reduce errors caused by the noise in single-point magnetometer measurements, but they require vehicles to move a certain distance and obtain a data sequence before each matching process, so their real-time performance is not strong. Moreover, most geomagnetic matching algorithms, such as ICCP and MAGCOM, only apply rigid transformations such as translation and rotation to input trajectories [27]. When applied to geomagnetic/odometry integrated navigation, the performance is limited due to the influence of a significant deviation between the actual trajectory and the matched trajectory shape caused by random factors in the odometry data.

In order to address the issue of the algorithm’s sensitivity to noise and the decrease in accuracy over time due to accumulated errors, this paper proposes a geomagnetic/odometry integrated localization method based on a real-time sequential particle filter (RSPF) for differential robot navigation. The main contributions of this paper can be summarized as follows:(1)Considering the additional errors caused by the influence of noise when the single-point geomagnetic particle filter algorithm operates continuously, we perform real-time sequential particle filtering by modifying the particles from single-point to first-in-first-out (FIFO) sequence using the data sequence from a segment of the trajectory. The particle weights are calculated using the data from the entire sequence, reducing the impact of measurement noise from individual points.(2)To minimize the positioning error caused by sequence matching based on rigid transformation when there is a substantial difference between the actual trajectory and the odometry trajectory, we incorporate the odometry calibration parameters of a differential robot into particles. The shape of the odometry trajectory is adjusted in real time, making it closer to the real trajectory.(3)To further improve the positioning accuracy, secondary matching of the matching results through the MAGCOM algorithm is performed to reduce the positioning errors of the sequential particle filter.

The rest of this paper is organized as follows. Section 2 describes the framework and design process of the RSPF-based localization method. In Section 3, simulation tests are given to verify the feasibility of the method. Section 4 proposes the experimental results and discussion. In Section 5, our works are concluded.

## 2. Proposed Method

The framework of the RSPF-based localization method is shown in Figure 1. In this section, we analyze the three components of our method, including RSPF, trajectory shape correction using odometry calibration parameters, and matching result correction using MAGCOM. We also describe the steps and implementation details of the method.

### 2.1. RSPF

The single-point-based geomagnetic particle filter has high real-time performance. However, when the amplitude of the noise exceeds the amplitude of geomagnetic field variations, continuous particle filtering may result in additional errors and even cause matching failure [18], making it difficult to use for long-range positioning. To improve the matching accuracy, prior research proposes using a particle filter based on a segment of the path, and experimental results have demonstrated the effective improvement in localization precision achieved through a path-matching-based particle filter [28]. Nonetheless, the execution of the algorithm still requires waiting for data sequence collection, thereby compromising the real-time performance of the particle filtering. In contrast to previous studies, this paper shifts from single-point matching to real-time sequence matching, ensuring real-time performance while minimizing the impact of noise.

The *i*-th particle of the *k*-th execution of the particle filter $p_{k}^{i}$ is defined as (1).
(1)$$p_{k}^{i}=\left[\begin{array}{c} x^{i}\\ y^{i} \end{array}\right],\;\;\;\;(x^{i},y^{i})\in S$$
where $x^{i}$ and $y^{i}$ are used to calculate the trajectory associated with the particle.

As shown in Figure 2, $P_{k}^{i}$ is the trajectory corresponding to $p_{k}^{i}$, and Q is the trajectory to be matched. R represents the real trajectory. $P_{k1}^{i}$, $Q_{1}$, and $R_{1}$ are the starting points of $P_{k}^{i}$, Q, and R, respectively. S is the constraint region of $P_{k1}^{i}$ with $Q_{1}$ as its center, and is used to prevent the position of $P_{k1}^{i}$ from being too far from $Q_{1}$, which could lead to matching failure. $P_{k}^{i}$ can be calculated as (2).
(2)$$P_{k}^{i}=Q+\left[\begin{array}{c} x^{i}\\ y^{i} \end{array}\right]$$

To improve the matching effectiveness, we calculate the particle weights using the X and Z components of the magnetic field vector along with the scalar value F, which represents the magnitude of the geomagnetic field vector. Due to the fact that the Y component can be calculated through X, Z, and F, ignoring the Y component has little effect on the feature dimension used for positioning. In addition, the calculation of geomagnetic three-axis vector data requires the attitude information of the vehicle [19], but the measurement of attitude sensors inevitably comes with noise, so the accuracy of geomagnetic vector data is often lower than that of scalar data. Therefore, we choose to ignore the Y component. Based on the particle weight formula proposed in reference [18], we define the weight of $p_{k}^{i}$ as (3).
(3)$$w_{k}^{i}=w_{k-1}^{i}\frac{1}{\sqrt{2\pi }}\exp(-\frac{\sum_{j=1}^{N_{D}}{\left(F_{real}^{j}-F_{mes}^{j}\right)}^{2}+\lambda {\left(X_{real}^{j}-X_{mes}^{j}\right)}^{2}+\lambda {\left(Z_{real}^{j}-Z_{mes}^{j}\right)}^{2}}{\tau N_{D}})$$
where $w_{k}^{i}$ is the weight of $p_{k}^{i}$; $N_{D}$ represents the length of the trajectory; $F_{real}^{j}$, $X_{real}^{j}$, and $Z_{real}^{j}$, respectively, represent the scalar and X and Z component values of the geomagnetic field at the *j*-th position on the reference maps; $F_{mes}^{j}$, $X_{mes}^{j}$, and $Z_{mes}^{j}$, respectively, represent the measurement values of scalar and X and Z components; τ is a constant selected based on the variance of measurement errors to prevent the error magnitude from being too large. Due to the influence of attitude sensor errors, the accuracy of the three-axis vector component measurements of the geomagnetic field is often lower than that of the scalar. Therefore, λ is needed to adjust the weight of vector components.

After calculating the weights of all particles, the weights are normalized using (4).
(4)$$\hat{w_{k}^{i}}=\frac{w_{k}^{i}}{\sum_{i=1}^{N}w_{k}^{i}}$$

Then, the particles are resampled using the method described in reference [15]. The result of the particle filter $p_{k}$ is calculated as (5).
(5)$$p_{k}=\sum_{i=1}^{N}p_{k}^{i}{w_{k}^{i}}^{T}$$
where T refers to matrix transpose. It can be seen from (3) that the weight of each particle is affected by the geomagnetic measurement values on a segment of trajectory, mitigating the effect of measurement noise of individual points. However, similar to geomagnetic matching algorithms, sequential particle filtering also requires the processing of data sequences collected over a period of time, which can impact real-time performance. In the field of gravity-aided navigation, some related studies have proposed the utilization of FIFO sequences to store a segment of data, enabling the implementation of real-time ICCP algorithms [29,30]. This kind of algorithm achieves real-time sequence matching, greatly improving the real-time performance. Inspired by these studies, this paper applies FIFO sequences to geomagnetic particle filtering. Before the algorithm is executed, a pre-collected data sequence of length $N_{D}$ is needed. Afterward, each time a new data point is obtained, the pre-stored data sequence is updated to generate particles and achieve real-time sequential particle filtering. As shown in Figure 3, assuming it is the *k*-th execution of the particle filter, the last $N_{D}$ points of the data to be matched are formed into data sequence $D_{k}$, and the trajectory $P_{k}^{i}$, which corresponds to the *i*-th particle $p_{k}^{i}$, is generated based on it. After one execution of the particle filter, the *k*-th point $Q_{k}$ is removed from the sequence and a new data point $Q_{k+N_{D}}$ is added to form a new sequence $D_{k+1}$ for the next filtering.

### 2.2. Trajectory Shape Correction Using Odometry Calibration Parameters

The data obtained from the differential wheel odometer is severely affected by random factors, resulting in a substantial difference between the actual trajectory and the odometry trajectory, as shown in Figure 4. Due to the rigid transformation of geomagnetic sequence matching algorithms, such as ICCP and MAGCOM, the matching performance is limited when applied to odometry trajectory matching. As shown in Figure 5, O represents the odometry trajectory, R represents the real trajectory, and P represents the matching result. Due to the significant difference in shape between O and R, P and R are difficult to overlap, resulting in positioning errors. Therefore, the particles generated by RSPF are not rigid transformations of O, but trajectories with modified shapes.

The motion principle of differential robots is based on a differential drive, which means controlling the speed difference between two wheels to achieve direction control and turning. As shown in Figure 6, the motion of a differentially steered robot can be approximated as a circular motion in a short period. $O_{C}$ is the center of the circular motion, r is the distance from $O_{C}$ to the center of two driven wheels, d is the distance from the driven wheel to the center of the two wheels, θ is the orientation of the robot at the previous moment, dθ is the angle increment during this period, ds is the displacement of the robot during this period, $r_{L}$ and $r_{R}$ are the radii of the left and right wheels, and $\omega_{L}$ and $\omega_{R}$ are the angular velocities of the left and right wheels.

The motion model of the differential wheeled robot is as follows:(6)$$v_{L}=r_{L}\omega_{L}$$
(7)$$v_{R}=r_{R}\omega_{R}$$
(8)$$v=\frac{v_{L}+v_{R}}{2}$$
(9)$$\omega =\frac{v_{R}-v_{L}}{2d}$$
where $v_{L}$ and $v_{R}$ represent the linear velocities of the left and right wheels; v represents the robot’s velocity at the current moment; and ω represents the robot’s angular velocity at the current moment.

The position of the robot can be calculated as follows:(10)$$\left[\begin{array}{c} ds\\ d\theta \end{array}\right]=\left[\begin{array}{c} \int vdt\\ \int \omega dt \end{array}\right]=\left[\begin{array}{cc} \frac{r_{L}}{2} & \frac{r_{R}}{2}\\ -\frac{r_{L}}{2d} & \frac{r_{R}}{2d} \end{array}\right]\left[\begin{array}{c} \int \omega_{L}dt\\ \int \omega_{R}dt \end{array}\right]$$
(11)$$\left[\begin{array}{c} x_{new}\\ y_{new} \end{array}\right]=\left[\begin{array}{c} x_{old}\\ y_{old} \end{array}\right]+ds\left[\begin{array}{c} \cos(\theta +d\theta )\\ \sin(\theta +d\theta ) \end{array}\right]$$
where dt is the sampling interval; ($x_{old}$, $y_{old}$) is the position of the robot at the previous moment; and ($x_{new}$, $y_{new}$) is the position of the robot at the current moment.

In practical applications, the parameters d, $r_{L}$, and $r_{R}$ may have some differences from the specifications provided by the robot manufacturer, and therefore require pre-calibration before use [1]. The accuracy of calibration is influenced by a calibration algorithm and the data being used and is one of the main causes of odometry trajectory errors. Adjusting the above parameters appropriately can affect the shape of the trajectory and reduce errors. The influence of the changes in different parameters on the trajectory shape is shown in Figure 7. 

This paper adds parameters d, $r_{L}$, and $r_{R}$ into the particles, allowing the trajectory shape of each particle to be adjusted. By employing geomagnetic particle filtering, the method generates results that closely resemble the real trajectory. The calculation of the particle $p_{k}^{i}$ can be modified as (12).
(12)$$p_{k}^{i}=\left[\begin{array}{l} r_{L}^{i}\\ r_{R}^{i}\\ d^{i}\\ x^{i}\\ y^{i} \end{array}\right]$$
where $r_{L}^{i}$, $r_{R}^{i}$, and $d^{i}$ are used to adjust the shape of the trajectory $P_{k}^{i}$ corresponding to the particle $p_{k}^{i}$, and $x^{i}$ and $y^{i}$ are used to add an overall offset to $P_{k}^{i}$.

Trajectory $P_{k}^{i}$ is shown in Figure 8. $P_{kj}^{i}$ is the *j*-th point on the trajectory. Assuming that θ is the initial orientation of the robot at the starting point of the trajectory, the calculation of the orientation of the robot at $P_{kj}^{i}$ is as (13).

(13)$$\left\{ \begin{array}{l} \theta_{j}=\theta ,\;\;\;\;\;j=0\\ \theta_{j}=\theta_{j-1}+\frac{\int (r_{R}^{i}\omega_{Rj}-r_{L}^{i}\omega_{Lj})dt}{2d^{i}},\;\;\;\;\;j>0 \end{array}\right.$$
where $\theta_{j}$ is the orientation of the robot at $P_{kj}^{i}$, $\theta_{j-1}$ represents the orientation at the previous point, and $\omega_{Lj}$ and $\omega_{Rj}$ are the angular velocities of the left and right wheels corresponding to $P_{kj}^{i}$. 

$P_{kj}^{i}$ can be calculated as (14).
(14)$$\left\{ \begin{array}{l} P_{kj}^{i}=O_{1}+\left[\begin{array}{c} x^{i}\\ y^{i} \end{array}\right],\;\;\;\;\;j=0\\ P_{kj}^{i}=P_{kj-1}^{i}+\frac{\int (r_{L}^{i}\omega_{Lj}+r_{R}^{i}\omega_{Rj})dt}{2}\left[\begin{array}{c} \cos(\theta_{j})\\ \sin(\theta_{j}) \end{array}\right],\;\;\;\;\;j>0 \end{array}\right.$$
where $P_{kj-1}^{i}$ is the previous point on $P_{k}^{i}$ and $O_{1}$ is the starting point of the odometry trajectory O.

Before performing particle filtering, it is necessary to pre-calibrate parameters $r_{L}$, $r_{R}$ and d of the odometer using a set of data to minimize the differences between the odometer trajectory and the actual trajectory. The pre-calibrated parameters are denoted as ${r_{L}}^{\prime }$, ${r_{R}}^{\prime }$, and $d^{\prime }$, which are used to generate odometer trajectories. In the process of particle filtering, it is necessary to limit the range of each particle to reduce the possibility of particle filter divergence. The range of $r_{L}^{i}$ is (${r_{L}}^{\prime }$-$\mu_{r_{L}}$, ${r_{L}}^{\prime }$+$\mu_{r_{L}}$), the range of $r_{R}^{i}$ is (${r_{R}}^{\prime }$-$\mu_{r_{R}}$, ${r_{R}}^{\prime }$+$\mu_{r_{R}}$), and the range of $d^{i}$ is ($d^{\prime }$-$\mu_{d}$, $d^{\prime }$+$\mu_{d}$).

### 2.3. Matching Result Correction Using MAGCOM

As shown in Figure 8, due to the imperfect match between $P_{k}^{i}$ and the real trajectory, there may be position errors in the positioning results. Meanwhile, the method may encounter matching failures in areas with a high noise impact on the geomagnetic data. Therefore, a subsequent correction of the matched results is needed. The MAGCOM algorithm works by performing a translational transformation on the trajectory to be matched, iterating over nearby grid points, and using mean square difference (MSD) to obtain the position with the lowest error in terms of geomagnetic field intensity. This algorithm exhibits fast computation, but it is highly influenced by the similarity of the trajectory shapes. Given that RSPF can effectively reduce trajectory shape errors, we choose to use MAGCOM as the method for correcting matching results. After executing RSPF for a certain distance, we use MAGCOM to perform secondary matching on the preliminary matching results.

During the execution of MAGCOM, the search points that may serve as matching results are commonly selected as $A_{1}\times B_{1}$ grid points on the geomagnetic reference map near the trajectory to be matched. Since the position error may be smaller than the grid length $L_{1}$, this paper divides grids in the search area into sub-grids of length $L_{2}$. The data of the sub-grid points are obtained through bilinear interpolation. The search points are selected as the nearest $A_{2}\times B_{2}$ sub-grid points to the starting point of the trajectory to be matched. As shown in Figure 9, O is the trajectory composed of the last M points in the particle filter matching result sequence $R_{s}$, and R is the corresponding true trajectory. $P^{i}$ is the *i*-th search trajectory of MAGCOM. Considering the noise impact of geomagnetic vector data, the MSD calculation is as follows:(15)$$e_{MSD}^{i}=\frac{1}{M}\sum_{j=1}^{M}{\left(F_{i}^{j}-F_{mes}^{j}\right)}^{2}+\gamma {\left(X_{i}^{j}-X_{mes}^{j}\right)}^{2}+\gamma {\left(Z_{i}^{j}-Z_{mes}^{j}\right)}^{2}$$
where $e_{MSD}^{i}$ is the MSD of $P^{i}$; $F_{i}^{j}$, $X_{i}^{j}$, and $Z_{i}^{j}$, respectively, represent the scalar and X and Z component values of the geomagnetic field at the *j*-th position of $P^{i}$ on the reference maps; $F_{mes}^{j}$, $X_{mes}^{j}$, and $Z_{mes}^{j}$, respectively, represent the measurement values of the scalar and X and Z components. γ is a constant used to adjust the weight of vector components due to the lower accuracy of the geomagnetic field vector component measurements than that of the scalar.

To ensure the real-time performance of the method, we only replace the endpoint of $R_{s}$ with the endpoint of the result of MAGCOM after each correction. When executing RSPF next time, an offset will be added to all positions in data sequence $D_{k}$. The offset can be calculated as (16).
(16)$$\left[\begin{array}{c} {x^{\prime }}_{D_{k}}\\ {y^{\prime }}_{D_{k}} \end{array}\right]=\left[\begin{array}{c} x_{D_{k}}\\ y_{D_{k}} \end{array}\right]+\vec{R_{s\_last}P_{MAG\_last}}$$
where $R_{s\_last}$ is the endpoint of $R_{s}$; $P_{MAG\_last}$ is the endpoint of the result of MAGCOM; ($x_{D_{k}}$, $y_{D_{k}}$) represents the position coordinates in $D_{k}$; and (${x^{\prime }}_{D_{k}}$, ${y^{\prime }}_{D_{k}}$) is the position coordinates with added offset.

### 2.4. Method Steps

The method proposed in this paper utilizes RSPF to effectively reduce the impact of magnetometer measurement noise. By incorporating odometry calibration parameters into particles, the trajectory shapes can be adjusted in real time to avoid errors caused by rigid transformation. Finally, MAGCOM is used to perform secondary matching on the preliminary matching results, further improving the positioning accuracy. The method flowchart is shown in Figure 10, and the method includes the following steps:
(1)If a stop command is received, navigation is considered finished. Otherwise, step two is performed.(2)Obtain new geomagnetic and odometry data, combine them into one data point, and add the data point to data sequence $D_{k}$. If the length of $D_{k}$ reaches $N_{D}$, set i=1, m=1, k=1 and perform the third step. Otherwise, repeat the second step.(3)If i<N, perform the third step. Otherwise, the sixth step is performed.(4)Initialize particle $p_{k}^{i}$ and calculate trajectory $P_{k}^{i}$.(5)Calculate particle weight $w_{k}^{i}$ and set i=i+1. Then, go back to the third step.(6)Normalize the particle weights and perform resampling. Then, calculate the particle filter result $p_{k}$. (7)Calculate result trajectory $P_{k}$ based on $p_{k}$. Save the endpoint of $P_{k}$ as the preliminary matching result in result sequence $R_{s}$. Then, set i=1, m=m+1.(8)If m<M, perform the 10th step. Otherwise, the ninth step is performed.(9)The last M points are combined into a sequence and used to perform MAGCOM matching. Then, replace $R_{s\_last}$ with $P_{MAG\_last}$.(10)Add offset to all positions in $D_{k}$. Then, set m=1.(11)Remove the starting point of $D_{k}$. Set k=k+1. Go back to the first step.

## 3. Simulation

In this section, the feasibility and localization performance of the RSPF-based localization method is evaluated through simulation. We first establish a simulation environment. Then, we analyze the feasibility of the method by measuring the influence of different parameter settings on the method’s performance. Finally, we conduct comparative experiments to compare the localization performance and accuracy of the algorithm and related algorithms.

### 3.1. Simulation Setup

In this section, we introduce the setup of our simulation, including the simulation platform, the construction of the simulation environment, the evaluation metrics, and the related method used for comparison.

#### 3.1.1. Simulation Platform

The configuration of the simulation platform used in this paper is shown in Table 1.

#### 3.1.2. Simulation Environment Construction

This paper built a simulation environment based on the differential robot motion model in Section 2.2, as shown in Figure 11. The simulation area is limited to 10 m  ×  10 m. The maximum linear speed of the left and right wheels of the robot is 2 m/s, and the acceleration is $1\;{m/s}^{2}$. The parameters used for the reference trajectory are set to $r_{L}=120\;\mathrm{mm}$, $r_{R}=120\;\mathrm{mm}$, and d=250 mm. By manually controlling the robot’s movement within the area, simulated reference trajectories and odometry trajectories are generated. The sampling interval is 0.25 s, and the total number of data points in the trajectory is 200–300.

We use three simulated geomagnetic reference maps generated through the addition of multiple random sine signals and mean filtering, including the X and Z components of the vector data and scalar data, as shown in Figure 12. The geomagnetic map is divided into 60×60 grids, and the maximum geomagnetic intensity differences between the X, Z, and scalar reference maps are 19,172.20 nT, 9999.99 nT, and 15,718.47 nT, respectively. The geomagnetic values corresponding to each position on the simulated trajectories are obtained by bilinear interpolation of the grid points of the reference map.

#### 3.1.3. Reference Methods

This paper combines a real-time ICCP with a vector ICCP [25] and designs a real-time vector ICCP as a comparative method. The vector ICCP performs ICCP using three magnetic reference maps to reduce errors caused by single component matching. Additionally, this paper selects an adaptive fission particle filter (AFPF) [18] as a comparison. This method uses adaptive particle fission and sampling to reduce particle degradation and impoverishment, and it inserts a distance interval during each execution of the particle filter to lower the influence of noise. Furthermore, this article also compares single-point-based geomagnetic particle filters.

#### 3.1.4. Evaluation Metrics

For evaluating the positioning accuracy of the method, we use root mean square error (RMSE), max error, and error at the end as evaluation metrics, which are calculated as follows:(17)$$e_{RMSE}=\sqrt{\frac{1}{R_{s}}\sum_{m=1}^{R_{s}}{\left(x_{m}-\hat{x_{m}}\right)}^{2}+{\left(y_{m}-\hat{y_{m}}\right)}^{2}}$$
(18)$$e_{max}=\max(\sqrt{{\left(x_{m}-\hat{x_{m}}\right)}^{2}+{\left(y_{m}-\hat{y_{m}}\right)}^{2}})$$
(19)$$e_{end}=\sqrt{{\left(x_{R_{s}}-\hat{x_{R_{s}}}\right)}^{2}+{\left(y_{R_{s}}-\hat{y_{R_{s}}}\right)}^{2}}$$
where $e_{RMSE}$, $e_{max}$, and $e_{end}$ represent the values of RMSE, max error, and error at the end, respectively; $R_{s}$ represents the length of the matching result sequence; ($\hat{x_{m}}$, $\hat{y_{m}}$) denotes the *m*-th positioning result; ($x_{m}$, $y_{m}$) denotes the *m*-th reference position; ($\hat{x_{R_{s}}}$, $\hat{y_{R_{s}}}$) represents the endpoint position of the positioning result; and ($x_{R_{s}}$, $y_{R_{s}}$) represents the endpoint position of the reference trajectory.

For the execution efficiency evaluation, we use the average processing time for 100 data points to evaluate the execution efficiency, which is calculated as (20).
(20)$$t_{100}=100t_{R_{s}}/R_{s}$$
where $t_{100}$ is the average processing time for 100 data points and $t_{R_{s}}$ is the total processing time for all points.

### 3.2. Feasibility Evaluation

The feasibility of the method was evaluated. We first use the average RMSE to analyze the influence of different sequence lengths ($N_{D}$) on the robustness of the trajectory shape differences and noise effects. Then, we evaluate the execution efficiency under different $N_{D}$ using the average processing time for 100 data points. 

The other parameter settings for the method are as follows. The particle numbers N is set to 300. The range limit of the parameter is set to $\mu_{r_{L}}$ = 30 mm, $\mu_{r_{R}}$ = 30 mm, and $\mu_{d}$ = 50 mm to allow particles to cover the correct results, as much as possible, when the true calibration error of the odometer is unknown. The radius of S is set to 10 mm. The MAGCOM matching trajectory length M is 30. The length of the sub-grid is set to $L_{2}\;=\;0.05L_{1}$ and the search scope is 10 × 10 sub-grids. The weight constants are set to τ = 100, λ = 0.5, and γ = 0.5.

#### 3.2.1. Robustness Evaluation

In order to evaluate the robustness of the difference in trajectory shape, this paper selected 100 trajectories and conducted three sets of comparative experiments with different pre-calibrated parameters (${r_{L}}^{\prime }$, ${r_{R}}^{\prime }$ and $d^{\prime }$). The parameter selection is shown in Table 2. The greater the difference between the pre-calibrated parameters and $r_{L}$, $r_{R}$, and d, the greater the difference in trajectory shapes. Gaussian noise with a mean of zero is added to the data. The noise standard deviation for the X and Z component data is 50 nT, and the noise standard deviation for the scalar data is 100 nT.

The results are shown in Figure 13. It can be seen from the figure that when the value of $N_{D}$ is small, the difference in trajectory shape has a significant impact on the positioning effect. The larger the trajectory difference, the greater the positioning error. As the value of $N_{D}$ increases, the positioning error decreases. When $N_{D}$ reaches 10, the downward trend of all curves becomes flat. At this point, we believe that the method has strong robustness on trajectory shape difference.

To evaluate the robustness of the impact of noise, this paper selected 100 trajectories and conducted three sets of comparative experiments using data with different levels of Gaussian noise. The noise levels differ in terms of their standard deviations, with a mean of zero. The noise standard deviation for the X and Z component data is denoted as $\sigma_{v}$, and the noise standard deviation for the scalar data is denoted as σ. As mentioned in Section 2.1, due to the influence of measurement errors from the attitude sensors, the noise in the magnetic vector components is generally greater than the noise in the magnetic scalar. Therefore, we set $\sigma_{v}\;=\;2\sigma$. The selection of standard deviation is shown in Table 3 The pre-calibrated parameters used for generating the odometry trajectory are set to ${r_{L}}^{\prime }$ = 118 mm, ${r_{R}}^{\prime }$ = 120 mm, and $d^{\prime }$ = 495 mm.

The results are shown in Figure 14. It can be seen from the figure that as the value of $N_{D}$ increases, the positioning error decreases. The decreasing trend of the errors becomes flat when $N_{D}$ reaches seven. After this point, the positioning effect does not improve significantly. At this point, we believe that the method has strong robustness on the impact of noise.

#### 3.2.2. Efficiency Evaluation

To measure the execution efficiency, we use the 
average *t*_100_ taken from the 
aforementioned simulations.

The results are shown in Figure 15. From the figure, it can be seen that the increase in $N_{D}$ leads to an increase in computation time. For every increase of one in the value of $N_{D}$, the average $t_{100}$ approximately increases by 0.60 s. Considering the robustness and execution efficiency of the method, we choose $N_{D}$ = 8 as the optimal parameter.

### 3.3. Performance Evaluation

Based on the optimal parameters in Section 3.2, we evaluate the performance of the RSPF-based localization method through comparative experiments with related algorithms. 

The parameter settings for the methods used in 
this simulation are as follows. For the real-time ICCP, the sequence length is 
10, with a maximum iteration count of 20 and an iteration termination threshold 
of 100 mm. For the single-point particle filter, the particle number is 300, 
with a maximum initial distance for particle positions set at 800 mm. For AFPF, 
the particle number is 300, with an interval for each execution set at 1200 mm 
and a maximum initial distance for particle positions set at 800 mm. As for the 
RSPF-based localization method, *N_D_* is 
set to 8, and the remaining parameters are the same as in Section 3.2.

#### 3.3.1. Positioning Accuracy Evaluation

In order to evaluate the influence of different trajectory shapes and noises on the positioning accuracy, we select 100 trajectories and set different ${r_{L}}^{\prime }$, ${r_{R}}^{\prime }$, $d^{\prime }$, σ, and $\sigma_{v}$ for five comparative experiments, as shown in Table 4. The trajectories and positioning errors of the simulation results are shown in Figure 16, and the simulation metrics are presented in Table 5.

From Figure 16 and Table 5, it is illustrated that in simulation 1, when the odometry trajectory shape is close to the reference trajectory and there is no noise in the geomagnetic data, all algorithms perform well in positioning. Compared to the odometry trajectory, the real-time ICCP reduces the average $e_{RMSE}$ and $e_{end}$ by 19.43% and 29.78%, respectively, but there is a phenomenon of matching failure, which leads to an increase in $e_{max}$. Single-point particle filter reduces the average $e_{RMSE}$, $e_{max}$, and $e_{end}$ by 38.24%, 33.05%, and 23.94%, respectively. AFPF reduces three types of errors by 38.19%, 26.93%, and 10.90%, respectively. The RSPF-based localization method has the best localization accuracy, with errors reduced by 71.83%, 64.52%, and 76.37%, respectively.

In simulations 2 and 3, as the differences in trajectory shapes increased, the positioning performance of each algorithm decreased. The real-time ICCP has a suppressing effect on $e_{RMSE}$, but matching failures become more severe, leading to an increase in $e_{max}$ and $e_{end}$. The single-point particle filter is affected by noise, has a limited matching effect, and has good performance in the initial stage of localization, but there is divergence in the latter half of the trajectory, resulting in a limited positioning effect and an increase in $e_{max}$. AFPF has a weak ability to correct trajectory shapes, causing divergence in the early stage of positioning, resulting in an increase in all other errors except for $e_{RMSE}$ in simulation 3. On the other hand, the RSPF-based localization method can effectively correct trajectory shape differences and still exhibits good suppressing capabilities on all three types of errors.

In simulations 4 and 5, as the noise intensity increases, the positioning performance of each algorithm decreases. The real-time ICCP matches through contour lines and is sensitive to noise, resulting in a significant increase in $e_{RMSE}$ and $e_{max}$, and due to severe oscillations in the matching trajectory, $e_{end}$ is unstable and shows a decrease. The single-point particle filter has a good positioning effect in the early stage of the trajectory, but there is divergence in the latter half of the trajectory, resulting in a limited positioning effect and an increase in $e_{max}$. AFPF has good resistance to noise impact and a better matching effect than the single-point particle filter and can reduce three types of errors. The RSPF-based localization method has the best robustness against noise effects and still exhibits good suppressing capabilities on all three types of errors.

#### 3.3.2. Execution Efficiency Evaluation

In order to evaluate the execution efficiency of algorithms, this paper measures the average execution time of each algorithm in the above simulations. The results are shown in Table 6.

As can be seen from the table, the processing time of the real-time ICCP for 100 points is approximately 28.08 s, while the single-point particle filter, AFPF, and RSPF-based localization method take 3.35%, 0.96%, and 28.67% of its time, respectively. The real-time ICCP uses sequence matching, requires multiple iterations, has a large computational workload, and is the slowest in terms of calculation speed. The single-point particle filter only performs matching on a single point, resulting in a shorter processing time. The AFPF, on the other hand, incorporates particle adaptive fission, which increases the computational workload of a single particle filter. However, there is a distance interval between two matching processes in this algorithm, reducing the overall execution time. The computation time is only 28.72% of that of the single-point geomagnetic particle filter. The RSPF-based localization method employs a sequence of length 8 for particle filtering and requires an execution every time a new data point is obtained. Compared to single-point geomagnetic particle filtering, it increases the computational workload by approximately 8.56 times.

## 4. Experiments

The effectiveness of the method proposed in this article under ideal conditions has been validated through simulation experiments. To further demonstrate the practicality of the algorithm, we conduct experiments using a real differential robot and evaluate the performance of the proposed algorithm. We also compare our method with other related algorithms.

### 4.1. Experiment Environment

The experimental equipment is shown in Figure 17. The robot is equipped with a three-axis magnetometer, a WT901C attitude sensor (manufactured by Witmotion Company in Shenzhen, China), and wheeled odometers. The real trajectory of the robot is collected through the FZ Motion optical motion capture system. Using a set of measured location data from FZ Motion to calibrate the odometer, the robot parameters were obtained as ${r_{L}}^{\prime }$ = 148.32 mm, ${r_{R}}^{\prime }$ = 143.47 mm, and $d^{\prime }$ = 484.70 mm. The robot is controlled to move within the area through a remote control while collecting data with a sampling interval of 0.25 s. The total number of trajectory data points is 500–600. The experimental area is limited to 4.85 m × 6.22 m. The experiment used three reference maps of the measured real geomagnetic vectors X and Z, as shown in Figure 18. The geomagnetic map is divided into 60 × 60 grids, and the maximum magnetic intensity difference between the X, Z, and scalar reference map is 7896.88 nT, 18,473.76 nT, and 18,177.06 nT, respectively.

The comparison algorithms and evaluation metrics used in the experiment are the same as those in the simulation.

Some parameters of the methods used in the experiment are adjusted. For the single-point particle filter, the maximum initial distance for particle positions is set at 500 mm. For the AFPF, the interval for each execution is set at 800 mm, and the maximum initial distance for particle positions is 500 mm. For the RSPF-based localization method, we set λ = 0.3 and γ = 0.3, and $N_{D}$ is set to 10 to achieve better robustness. Other parameters are the same as the simulation.

### 4.2. Experimental Results and Performance Evaluation

This paper conducts matching experiments on 10 sets of real trajectories and presents a comparative analysis of the performance of the proposed method and related methods.

#### 4.2.1. Experimental Results

The trajectory and positioning errors of some experimental results are shown in Figure 19, and the statistical data is shown in Table 7 based on the defined evaluation metrics.

#### 4.2.2. Positioning Accuracy Evaluation

As shown in Figure 19 and Table 7, the proposed method in this paper achieves a higher level of positioning accuracy compared to other algorithms. Specifically, the real-time ICCP has a reduced average $e_{RMSE}$ and an average $e_{end}$ of 6.54% and 28.00%, respectively, while the average $e_{max}$ has shown a 16.60% improvement. The single-point particle filter has a reduced average $e_{max}$ by 4.74%, while the average $e_{RMSE}$ and the average $e_{end}$ have increased by 5.33% and 20.63%, respectively. The AFPF has a reduced average $e_{RMSE}$, average $e_{max}$, and average $e_{end}$ of 7.60%, 6.56%, and 8.68%, respectively. And the RSPF-based localization method has a reduced average $e_{RMSE}$, average $e_{max}$, and average $e_{end}$ of 34.04%, 41.23%, and 28.55%, respectively.

Due to the significant differences in shape between the odometry trajectory and the actual trajectory and the presence of high noise levels in the magnetometer data, both the real-time ICCP and the single-point particle filter experience a considerable number of matching failures, resulting in additional errors. On the other hand, AFPF, with an execution interval of 800 mm, experiences fewer matching failures. However, the positioning accuracy of AFPF is heavily influenced by trajectory shapes. As the robot’s traveling distance increases, the positioning performance gradually deteriorates, limiting its ability to reduce errors. In contrast, the method proposed in this paper demonstrates a good ability to correct trajectory shapes and exhibits robustness against noise, effectively suppressing odometry cumulative errors and achieving higher positioning accuracy.

#### 4.2.3. Efficiency Evaluation

In terms of execution efficiency, according to Table 7, the average processing time of the real-time ICCP for 100 points is approximately 20.77 s. The processing times of the single-point particle filter, AFPF, and RSPF-based localization method are 4.29%, 1.16%, and 40.63%, respectively, compared to the real-time ICCP. From Table 6 and Table 7, it can be observed that, compared to the simulated environment, the experimental area under real conditions is smaller, resulting in a decrease in the computational complexity of contour lines. As a result, the processing time of the real-time ICCP is reduced by 26.03% compared to the simulation. However, it still has the longest processing time compared to other algorithms. The single-point particle filter still has the shortest computation time, followed by AFPF. Due to the increase of $N_{D}$ to 10, the time consumption of the RSPF-based localization method has increased to 9.48 times that of the single-point particle filter, but it is still faster compared to the real-time ICCP. Considering the improved accuracy, this level of computational efficiency is acceptable.

### 4.3. Discussion

The generality and efficiency of the RSPF-based localization method are discussed in this section:(1)Discussion of Generality:

RSPF can process data sequences in real time to reduce the impact of high noise levels in measurement data and improve the robustness of the localization algorithm. Simultaneously, when integrated with the odometry calibration model, it mitigates the influence of trajectory shapes, resulting in the achievement of high-precision positioning results. This strategy can be applied to other multi-sensor fusion localization algorithms based on motion models with severe noise in the data, including, but not limited to, geomagnetic/INS integrated navigation and others.

However, there are still some issues with our selection of range limit at present. In order to maintain the universality of the parameters, we have chosen larger constraint parameters, $\mu_{r_{L}}$, $\mu_{r_{R}}$, and $\mu_{d}$, which may lead to potential algorithm divergence or wastage of computing resources. We will explore more suitable parameter selection in our future research.

(2)Discussion of Efficiency:

The real-time analysis of a data sequence may introduce heightened computational complexity, and the computation time is approximately the product of the processing time of the single-point algorithm and sequence length, resulting in a decrease in execution efficiency. We will explore in future research how to lightweight algorithms to further improve real-time performance while maintaining positioning accuracy. 

(3)Discussion of Robustness:

Simulation results show that the RSPF-based localization method has strong robustness against zero mean Gaussian noise. However, compared to the simulation, the localization performance of the method has decreased in real environments, which may be due to the complexity of noise in real environments. We will consider how to reduce the noise impact in real environments in future research, such as adaptively adjusting sequence length and designing more suitable particle weight formulas.

## 5. Conclusions

In this paper, we proposed a geomagnetic/odometry integrated localization method based on RSPF for differential robot navigation. The proposed RSPF method used the data sequence from a segment of the trajectory to perform particle filtering. This approach reduced positioning errors caused by magnetometer measurement noise in a single-point particle filter while maintaining real-time performance. Additionally, the method incorporated the odometry calibration parameters of a differential robot to adjust the trajectory shapes, thereby mitigating errors introduced by rigid transformations applied to the trajectory. Lastly, secondary matching on the matching results through the MAGCOM algorithm was performed to reduce the potential position errors of the particle filter. The experimental results indicated that, compared to the odometry trajectory, the average $e_{RMSE}$, average $e_{max}$, and average $e_{end}$ have been reduced by 34.04%, 41.23%, and 28.55%, respectively. However, compared to the single-point particle filter, this algorithm will result in an increase in computational complexity and an average processing time of 9.48 times, which leads to higher hardware support when applied.

In summary, the proposed method can effectively improve positioning accuracy and offers an important reference to geomagnetic-aided localization in other applications. But further research is still needed to reduce the complexity of the method.

## Figures and Tables

**Figure 1 sensors-24-02120-f001:**
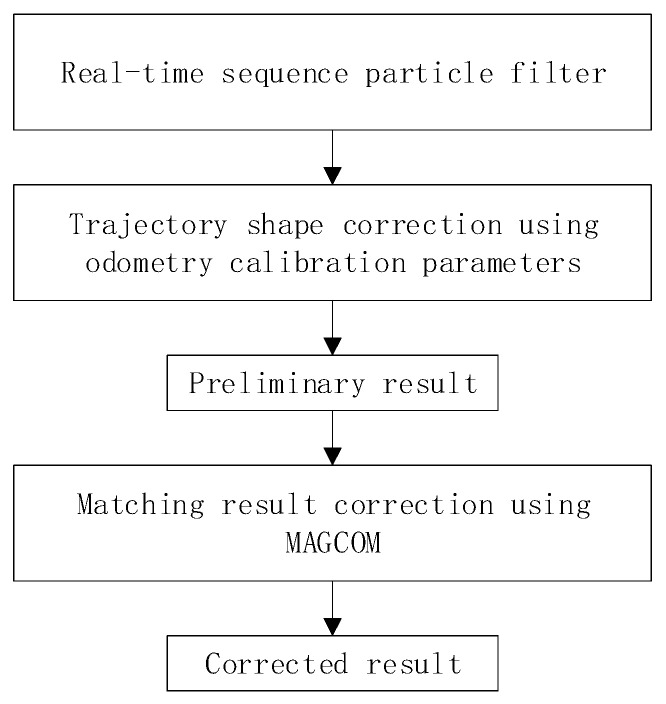
The framework of the proposed method.

**Figure 2 sensors-24-02120-f002:**
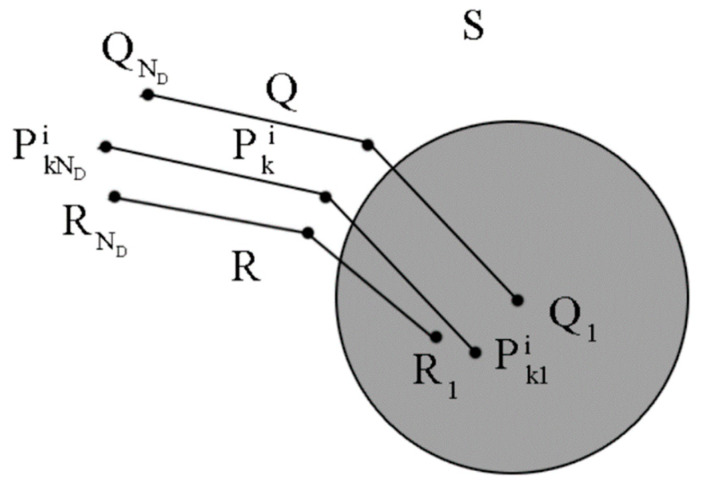
Sequential geomagnetic particle filter.

**Figure 3 sensors-24-02120-f003:**
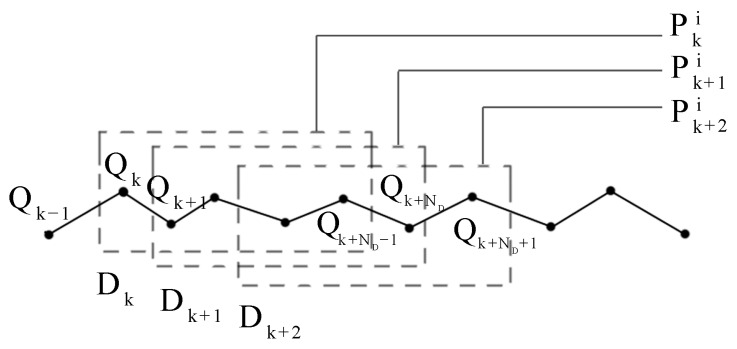
Real-time sequential particle filter based on FIFO.

**Figure 4 sensors-24-02120-f004:**
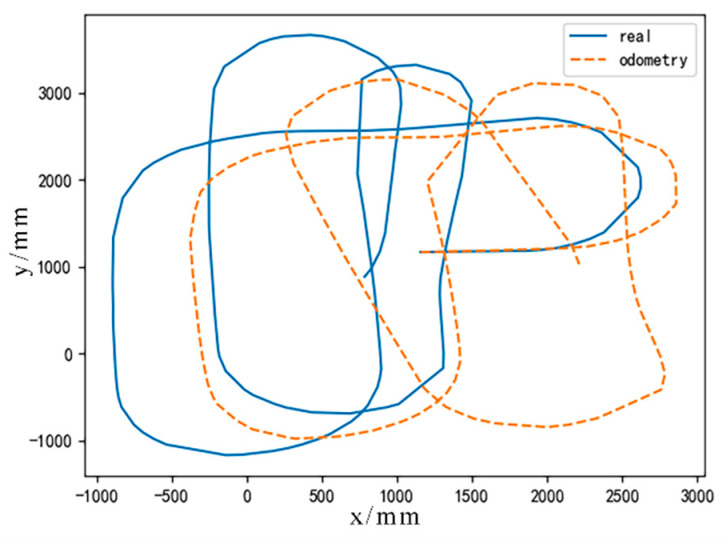
Comparison between real trajectory and odometry trajectory.

**Figure 5 sensors-24-02120-f005:**
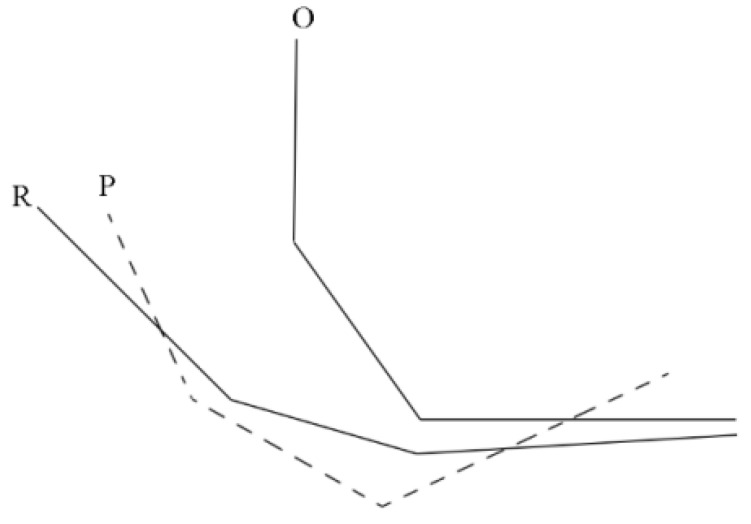
Rigid transformation result.

**Figure 6 sensors-24-02120-f006:**
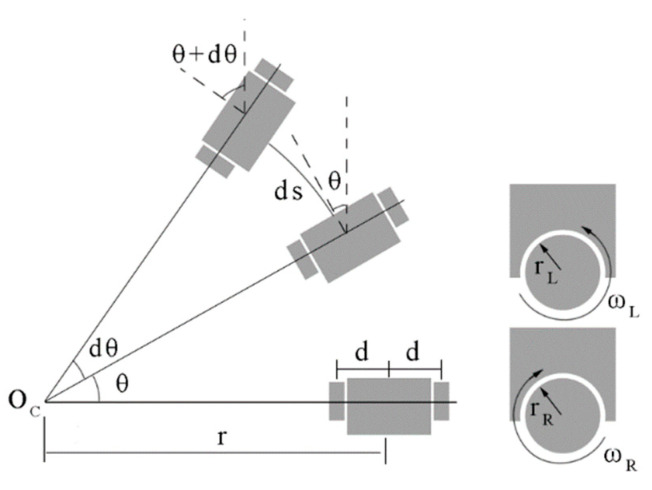
The motion of differential robots.

**Figure 7 sensors-24-02120-f007:**
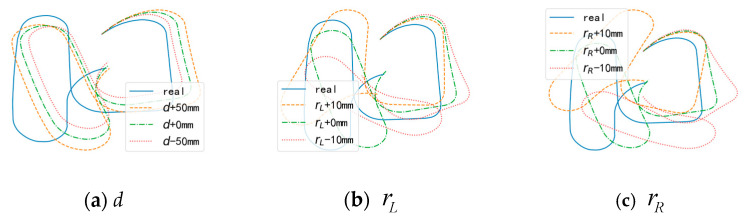
The influence of different odometry calibration parameters. (**a**) Trajectories generated by different values of d. (**b**) Trajectories generated by different values of $r_{L}$. (**c**) Trajectories generated by different values of $r_{R}$.

**Figure 8 sensors-24-02120-f008:**
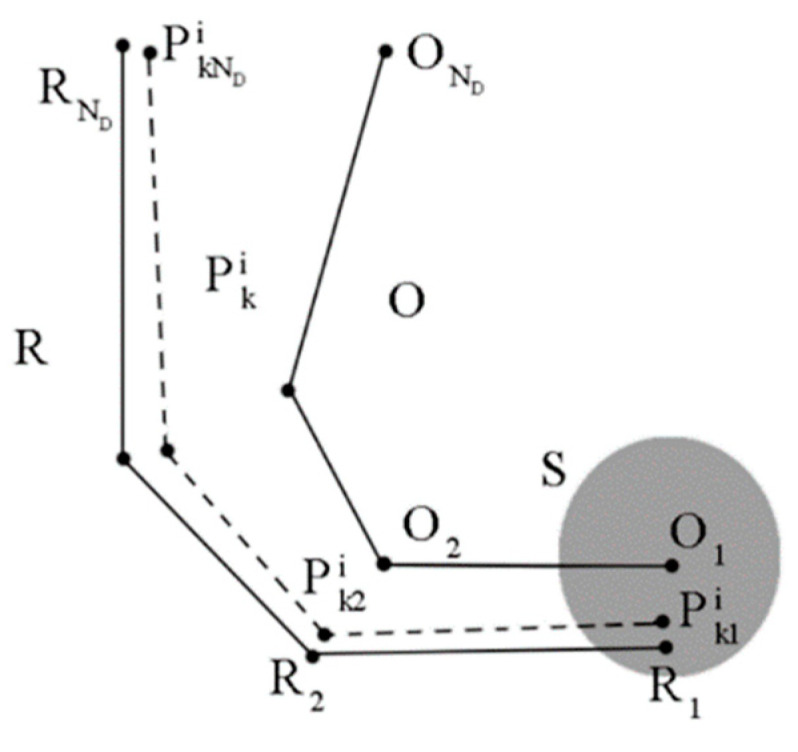
Trajectory shape correction.

**Figure 9 sensors-24-02120-f009:**
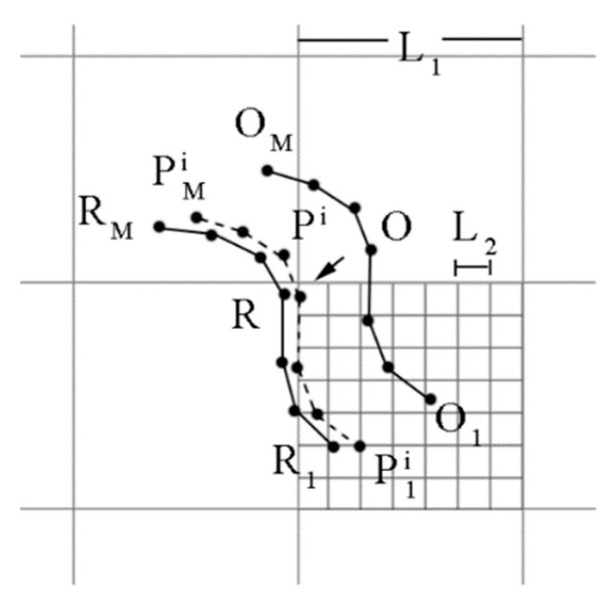
Matching result correction.

**Figure 10 sensors-24-02120-f010:**
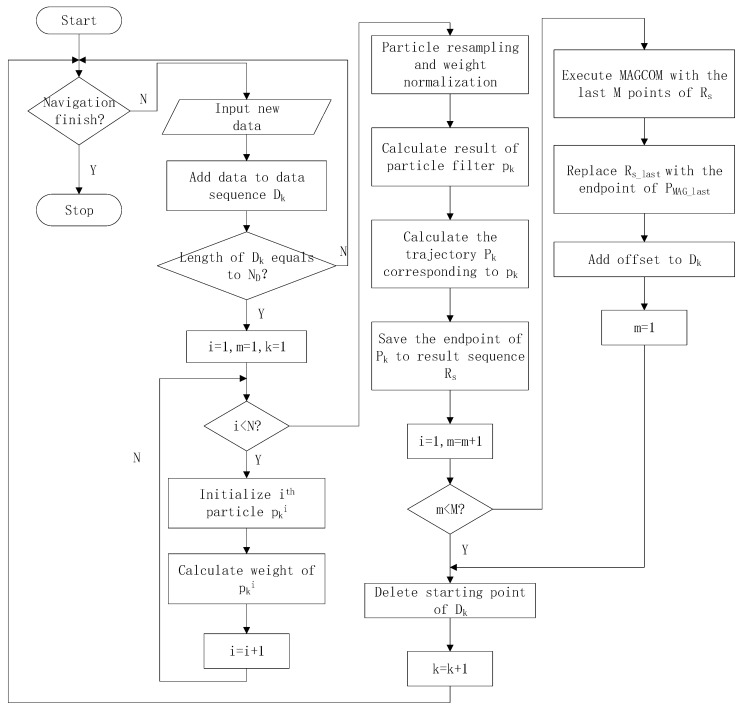
The flowchart of the RSPF-based localization method.

**Figure 11 sensors-24-02120-f011:**
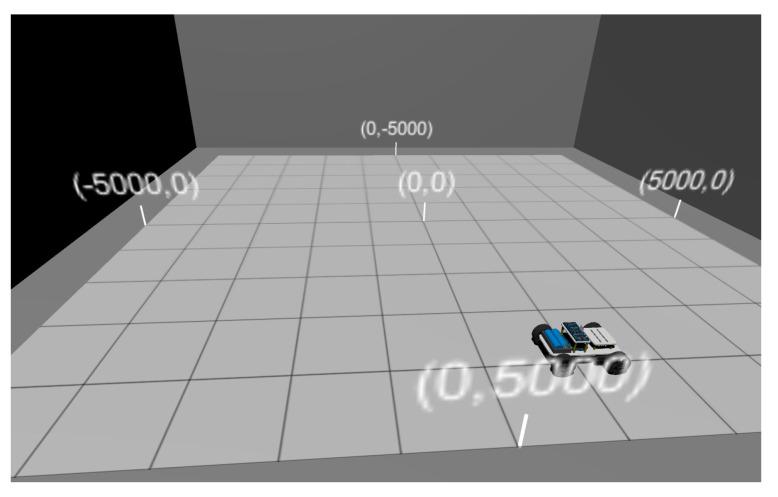
Robot motion simulation.

**Figure 12 sensors-24-02120-f012:**
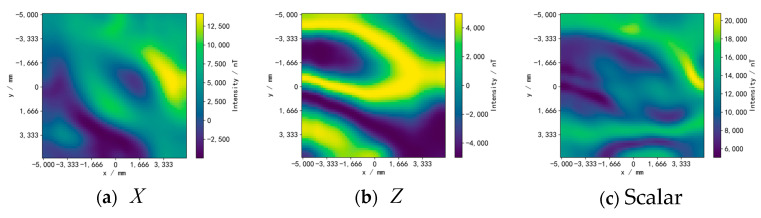
Simulated geomagnetic reference maps. (**a**) Reference map of the X component. (**b**) Reference map of the Z component. (**c**) Reference map of the scalar.

**Figure 13 sensors-24-02120-f013:**
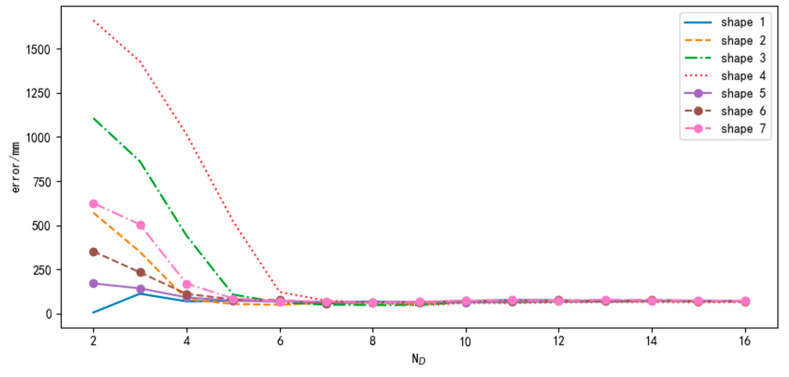
Average RMSE under different trajectory shapes.

**Figure 14 sensors-24-02120-f014:**
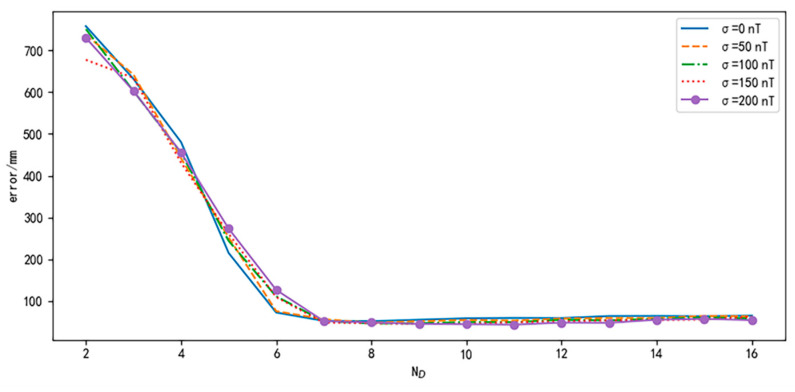
Simulation results under different noises.

**Figure 15 sensors-24-02120-f015:**
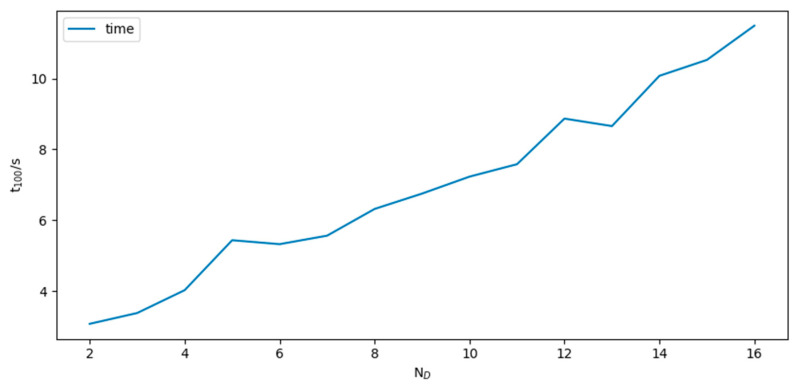
Average simulation time.

**Figure 16 sensors-24-02120-f016:**
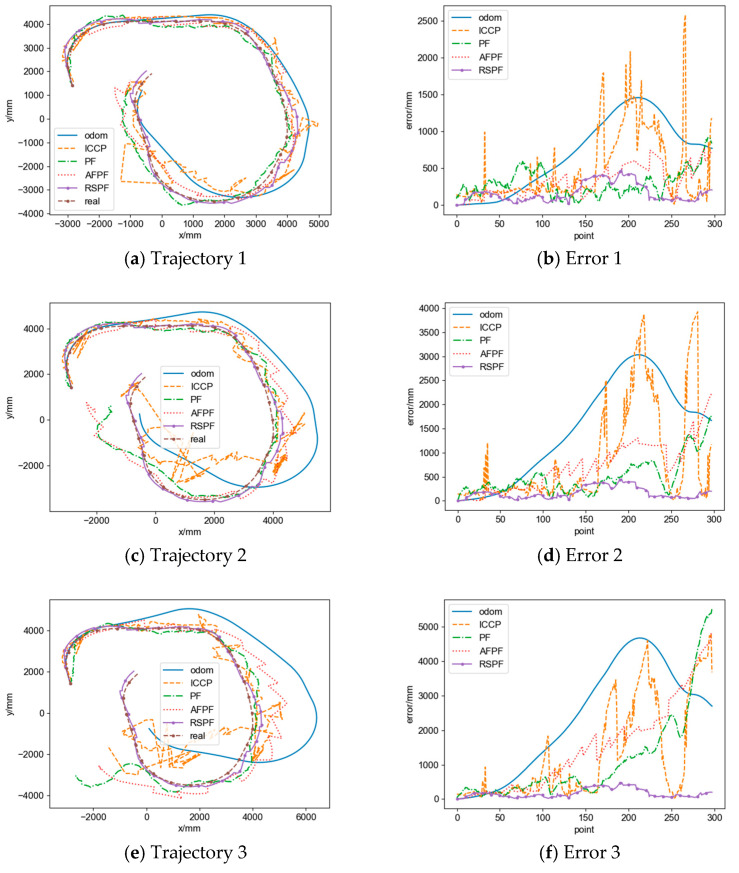
Comparison of trajectories and positioning errors of different simulations.

**Figure 17 sensors-24-02120-f017:**
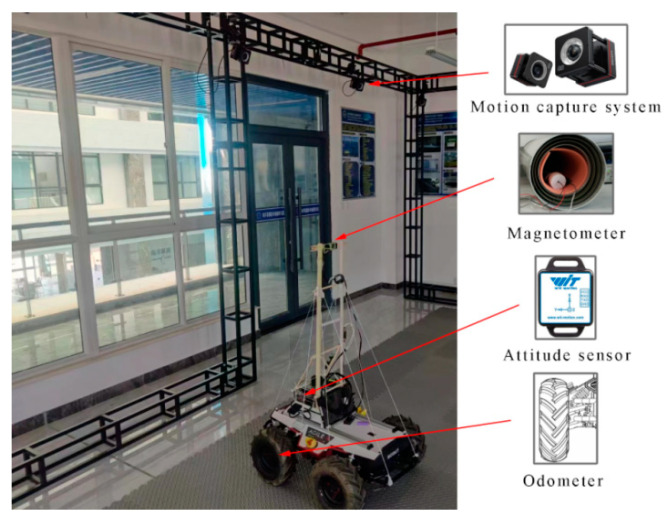
Experimental equipment.

**Figure 18 sensors-24-02120-f018:**

Geomagnetic reference maps. (**a**) Reference map of the X component. (**b**) Reference map of the Z component. (**c**) Reference map of the scalar.

**Figure 19 sensors-24-02120-f019:**
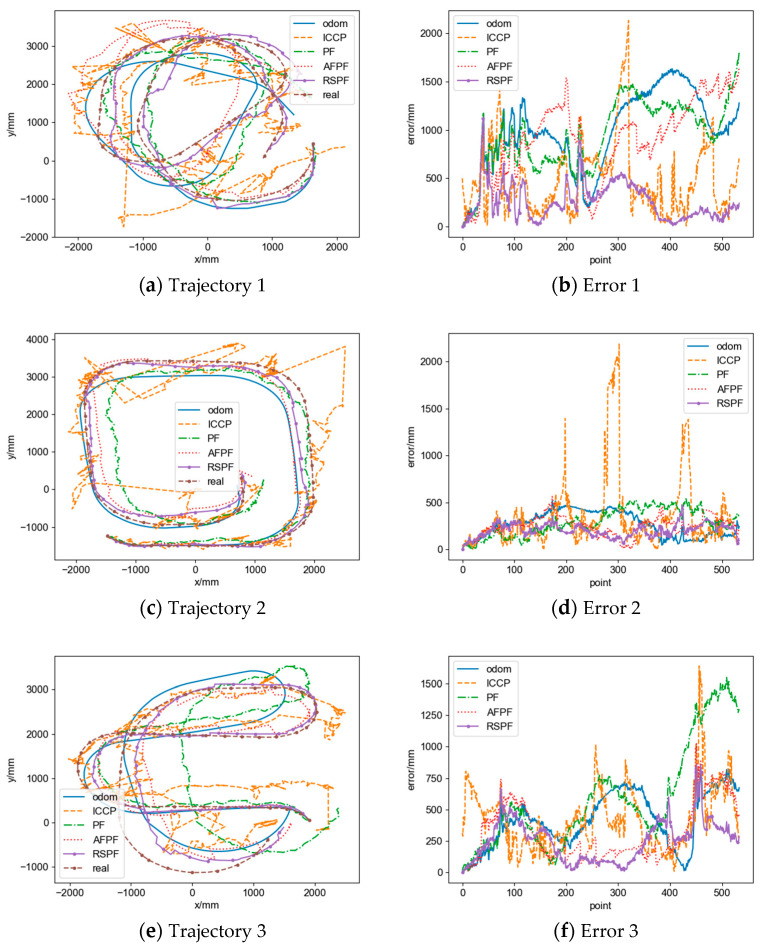
Trajectories and positioning errors of experimental results.

**Table 1 sensors-24-02120-t001:** Main configuration of the PC.

Components	Specifications
CPU	Intel(R) i7-10870H @ 2.20GHz
RAM	16 GB
Operating System	Windows 10 (64-bit)
Simulation Software	Unity 2019.3.3f1, PyCharm Community Edition 2023.2.1

**Table 2 sensors-24-02120-t002:** Pre-calibrated parameters.

Shape	${r_{L}}^{\prime }$/mm	${r_{R}}^{\prime }$/mm	d′/mm
1	120	120	500
2	119	120	500
3	118	120	500
4	117	120	500
5	120	120	495
6	120	120	490
7	120	120	485

**Table 3 sensors-24-02120-t003:** Noise standard deviations.

Magnetic Noise	σ/nT	$\sigma_{v}$/nT
1	0	0
2	50	100
3	100	200
4	150	300
5	200	400

**Table 4 sensors-24-02120-t004:** Simulation parameters for comparative experiments.

	${r_{L}}^{\prime }$/mm	${r_{R}}^{\prime }$/mm	d′/mm	σ/nT	$\sigma_{v}$/nT
1	119	120	495	0	0
2	118	120	490	0	0
3	117	120	485	0	0
4	119	120	495	50	100
5	119	120	495	100	200

**Table 5 sensors-24-02120-t005:** Comparison of accuracy of different methods.

		Odometer	Real-Time ICCP	PF	AFPF	RSPF
1	Average $e_{RMSE}$/mm	714.72	575.86	441.43	441.78	201.31
	Average $e_{max}$/mm	1458.34	3021.55	976.41	1065.65	517.33
	Average $e_{end}$/mm	1061.40	745.30	807.33	945.76	250.78
2	Average $e_{RMSE}$/mm	1489.94	1127.50	1275.24	1511.65	236.34
	Average $e_{max}$/mm	3032.14	3925.28	2229.72	3957.33	563.16
	Average $e_{end}$/mm	2169.55	1825.05	1941.41	2646.74	266.51
3	Average $e_{RMSE}$/mm	2291.13	1699.99	1641.35	1937.64	242.18
	Average $e_{max}$/mm	4669.68	4793.87	4544.77	5063.85	570.73
	Average $e_{end}$/mm	3248.70	3394.47	2276.20	3254.32	324.92
4	Average $e_{RMSE}$/mm	714.72	1218.29	501.73	413.06	210.40
	Average $e_{max}$/mm	1458.34	5640.38	1219.81	1025.42	538.13
	Average $e_{end}$/mm	1061.40	523.01	706.00	255.08	255.08
5	Average $e_{RMSE}$/mm	714.72	1456.84	636.10	447.93	210.87
	Average $e_{max}$/mm	1458.34	5994.75	1734.87	1055.51	562.40
	Average $e_{end}$/mm	1061.40	508.84	1194.82	906.51	274.50

**Table 6 sensors-24-02120-t006:** Comparison of efficiency of different methods.

	Real-Time ICCP	PF	AFPF	RSPF
Average $t_{100}$/s	28.08	0.94	0.27	8.05

**Table 7 sensors-24-02120-t007:** Average results of different algorithms on all experimental trajectories.

	Odometer	Real-Time ICCP	PF	AFPF	RSPF
Average $e_{RMSE}$/mm	556.76	520.33	586.45	514.13	367.19
Average $e_{max}$/mm	1884.59	2197.41	1795.29	1761.01	1107.43
Average $e_{end}$/mm	348.90	251.22	420.87	318.60	249.28
$t_{100}$/s	-	20.77	0.89	0.24	8.44

## Data Availability

The data that support the findings of the study are available from the corresponding author on reasonable request.

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
