# Peer review of "A Geomagnetic/Odometry Integrated Localization Method for Differential Robot Using Real-Time Sequential Particle Filter"

_sensors, 2024, doi:10.3390/s24072120_

Round 1
Reviewer 1 Report
Comments and Suggestions for Authors
The paper proposes corresponding improvement schemes and algorithms to address the potential problems of geomagnetic positioning algorithms based on single point particle filters and the significant impact of random factors on differential wheel odometry, thereby enhancing the performance of the positioning algorithm. The paper is logically clear and organized, making the entire paper smooth to read and easy to understand. In addition, the authors show excellent rigor in logical reasoning, which makes the proposed model and conclusion more convincing. The numerical simulation results show that the proposed method is effective, so it is recommended to be published.
However, there are still some shortcomings in the details of the paper, mainly in the following aspects:
1.The format of the paper is inappropriate or incorrect. For example, formulas (3), (15), etc. clearly go beyond the main text and require the author's confirmation;
2. Fig 7, Fig 12, Fig 13, and other images clearly exceed the main text and require the author's confirmation;
3. The symbols in
,
in formula (17), (18) and
in
,
in (19) are exactly the same, but they do not represent the same variable in lines 433, 434, and 435. The author's confirmation is required;
4. There is a blank line below formula (20) and the word "where" in the following text is bolded, which does not match the format of the text. In addition, the sentence does not end with punctuation, and the author needs to confirm and modify it.

Comments on the Quality of English Language
English writing expression needs to be improved.
Author Response
Dear Reviewer, Thank you for your thorough review of our manuscript and for providing valuable feedback. We have carefully considered each of your comments and suggestions, and we appreciate the time and effort you dedicated to evaluating our work. In response to your comments, we have made several revisions to the manuscript to address the issues raised and to improve the clarity and quality of our research. A detailed response to each of your points, along with the corresponding revisions, is provided in the attached PDF document. We hope that our responses adequately address your concerns and that you find the revisions satisfactory. We remain open to any further suggestions or clarifications you may have. Thank you once again for your valuable input and for helping us improve our manuscript. Sincerely, Xiaozhen Yan

Reviewer 2 Report
Comments and Suggestions for Authors
The proposed work is devoted to the issues of autonomous navigation of self-propelled objects in conditions where the use of signals from satellite navigation systems, such as NAVSTAR GPS, GLONASS, BEIDOU, GALILEO, is unavailable or difficult. This situation is typical for underwater vehicles, as well as for robots moving in underground structures and shelters where L-band radio waves do not penetrate. In this case, the use of odometers is assumed, i.e. devices that record the distance traveled and angles of rotation (more common for land robots) and inertial systems based on the use of accelerometers with subsequent integration to determine the speed and path traveled (more often used for underwater vehicles). Both systems are characterized by the accumulation of location errors due to inaccurate sensor operation and the presence of mechanical interference. In this regard, additional measures are used to clarify the position of the moving object. The advantages and disadvantages of these methods are analyzed in the proposed work. It is proposed to use geomagnetic field orientation as a method that significantly improves location accuracy. It is shown that in this case the accuracy of the algorithm can be significantly improved, but the requirements for computing resources increase. The result seems quite expected, but the experimental testing and comparison of simple and improved algorithms deserves attention. The article may be of methodological interest for developers of autonomous navigation systems for self-propelled robots. Perhaps the issues of obtaining a digital magnetic card for the test site should be covered.
Author Response
Dear Reviewer, Thank you for your thorough review of our manuscript and for providing valuable feedback. We have carefully considered each of your comments and suggestions, and we appreciate the time and effort you dedicated to evaluating our work. Sincerely, Xiaozhen Yan

Reviewer 3 Report
Comments and Suggestions for Authors
Summary
This paper integrates three established methods including wheel odometry, geomagnetic contour matching, and particle filtering to provide an improved navigation performance, specifically for differential robots in an indoor environment. Variations of the dimensional parameters are integrated in the algorithm allowing pre-calibration of the differential wheel odometry sensor. To minimize the effect of magnetic noise at a specific location, a fixed-length sequence of data was fed to the filtering algorithm, and the sequence was updated based on a first-in-first-out protocol to achieve rela-time performance. An auxiliary matching was performed on the geomagnetic data to further minimize the positioning errors. The performance of the presented model was tested through simulations and experiments using a real differential robot.
Overall, the manuscript is well presented and reads fluently except some ambiguous sentences or terms. Below are my general and specific comments that need to be addressed.
General Comments
The abstract requires quantitative performance metrics to substantiate the acclaimed “higher positioning accuracy”.
The introduction needs to introduce the “differential robots” and “particle filters” as they are the pillars of the work. The description of the “differential robots” (Lines 254-255) needs to be presented before “Wheel odometry” (Line 36) to avoid ambiguities.
Although section 2 adds to the concepts presented in lines 60-103, it resembles another introduction repeating part of the conveyed information. The authors are encouraged to merge these sections while maintaining the same titles as in sections 2.1 and 2.2. Specifically, merge section 2.1 with lines 60-68, section 2.2 with lines 69-82, and lines 166-176 with lines 83-103.
The units of the physical parameters are mostly neglected in the figures and parts of the paragraphs. Moreover, the authors are encouraged to include specific, descriptive captions for the figures and tables. Recycling the captions (as the authors did for Tables 2, 3, and 5) devalues the merits of the work. Inclusion of the term “schematic” in each figure caption seems redundant as the readers know what type of illustration is presented.
Section 5.2 needs to clarify that the experimental results presented in Table 8 were calculated for which of the trajectories. Is there any reason that Table 8 does not provide information for other trajectories?
In the conclusion section, numerically compare the latency of the proposed method with that of the single-point particle filter.
Specific Comments
Line 27: For MAGCOM, the term “Magnetic” is a better match than “Geomagnetic”. You may adopt the term Geomagnetic Matching Navigation (GMN) similar to reference [11].
Line 147: Define the acronym SIMAN here or in the “List of abbreviations” after the abstract.
Lines 167 & 188: The “long-range” positioning is ambiguous. Provide some quantitative metrics (in line 167) for short-range and long-range positioning.
Line 186: What level of magnetometer noise will be considered as “large” noise?
Line 196: There is a gap in the flow of the contents between lines 195 & 196. Conceptually define the “particle” and its significance.
Lines 199-200: Clarify that the trajectory O is obtained via the magnetometer data. Using the notation O for the trajectory “to be matched” creates ambiguities in this section as section 3.2 states that the trajectory O is obtained via odometry.
Line 201: Explain why the constraint S needs to be imposed.
Line 204:
Include “F” after “scalar value”.
What information does the scalar value convey (is it the magnitude of the X, Y, and Z components)?
What is the rationale for neglecting the Y component of the magnetic field.
Equation (3): Cite a source for this weight allocation model.
Lines 212-214: This argument is vague. Clarify if this concept is valid for any type of magnetometer, or it only applies to the specific sensor used in this study.
Line 217: Please clarify how the particles are resampled.
Line 218: Clarify if T in (5) refers to the “transpose” operation.
Line 219: Revise (2) to (3).
Lines 227-234: Explain that the first segment of the trajectory cannot be processed real-time until Np data points are collected, and the FIFO process starts.
Figure 2: The trajectory R was not defined in section 3.1.
Line 236: Use a proper caption including the term “geomagnetic particle filter” as stated in lines 227-228.
Figure 3: The assignment of the subscripts needs to be revised. If O1 is the “starting point” of the trajectory O, the first execution of the particle filter will be P1i which will be applied to the sequence D1.
Line 237: Present a descriptive caption for Figure 3 including the terms “sequential” and/or “FIFO”.
Line 242: Clarify what the “rigid transformation” is.
Line 248: Clarify which “proposed method”. Does it refer to the RSPF, ICCP, or MAGCOM?
Figure 4: Include the axes titles and units.
Figure 7: What are the units for numbers 10 and 50 in the legends?
Line 287: In the figure caption, use the term “odometer” or “odometry” to specify the parameters.
Equations (13) & (14): j ≠ 0 implies that negative values can be assigned. Use j > 0 or other appropriate notation.
Figure 8: Explain why the 1st trajectory segment of P is distinguished with a dashed line.
Line 330: Clarify what the search points are.
Line 364: Clarify which type of data is obtained. Is it the geomagnetic data or odometry?
Lines 365-366: There is a disagreement between this step and the flowchart. The statement here indicates inputting new data, whereas the flowchart returns to the Navigation Finish verification.
Lines 364-378: There is no indication of Navigation Finish verification and how it will be achieved.
Line 404: Explain what algorithm was used to generate the simulated geomagnetic reference maps.
Line 407: What is the unit for the presented geomagnetic data?
Figure 12: Include the axes titles and units. Also specify the unit for the magnetic data on the sidebars.
Equations (17), (18), and (19): The indices of x and y are identical for the squared terms resulting in 0 values. Please correct them in the equations and in lines 433-435.
Lines 449-450: The 30 mm and 50 mm values are excessive considering the numbers presented in Table 2 which shows maximum variations of 3 mm and 15 mm for r L’ and d’, respectively.
Lines 474-476: This vague statement was already mentioned in lines 212-214. Please refer to my comment for lines 212-214.
Tables 2, 3, and 5: Use specific caption for each table.
Figure 14: Include the unit for sigma values.
Line 486: Clarify that the noise is magnetic.
Figures 16 and 19: For readers convenience, I strongly recommend using the same line type for each algorithm in the “Trajectory” and “Error” plots (i.e., omit the solid blue line in the error plots).
Line 586: Clarify what type of “measured data”. Location data from FZ motion or magnetic data?
Line 593: Present the unit for the numbers.
Figure 18: Same as my comment for figure 12.
Comments on the Quality of English Language
English Usage
Line 122: Revise “is used” to “was used”.
Line 322: Suggestion, revise “translation” to “translational”.
Line 389: Revise “introduced” to “introduce”.
Line 418: Use capital letter V for “vector”.
Use the past tense in the conclusion section.
Line 683: Revise “propose” to “propose”.
Line 691: Revise “is” to “was”.
Line 692: Revise “indicate” to “indicated”.
Author Response

(The authors gave the same response as above.)
